# Applying Access Control Enabled Blockchain (ACE-BC) Framework to Manage Data Security in the CIS System

**DOI:** 10.3390/s23063020

**Published:** 2023-03-10

**Authors:** Abdullah Alharbi

**Affiliations:** Computer Science Department, Community College, King Saud University, Riyadh 11437, Saudi Arabia; arharbi@ksu.edu.sa

**Keywords:** cyber information sharing, blockchain, security, privacy, unauthorized users, access control enabled blockchain (ACE-BC)

## Abstract

Cybersecurity information sharing (CIS) is important in different business processes to secure data transmission, because it comprises Internet of Things (IoT) connectivity, workflow automation, collaboration, and communication. The shared information is influenced by intermediate users and alters the originality of the information. Although risk factors such as confidentiality and privacy of the data are reduced when using a cyber defense system, existing techniques rely on a centralized system that may be damaged during an accident. In addition, private information sharing faces rights issues when accessing sensitive information. The research issues influence trust, privacy, and security in a third-party environment. Therefore, this work uses the Access Control Enabled Blockchain (ACE-BC) framework to enhance overall data security in CIS. The ACE-BC framework uses attribute encryption techniques to manage data security, while the access control mechanism limits unauthorized user access. The effective utilization of blockchain techniques ensures overall data privacy and security. The efficiency of the introduced framework was evaluated using experimental results, and the experimental outcome indicated that the recommended ACE-BC framework enhanced the data confidentiality ratio (98.9%), the throughput ratio (98.2%), the efficiency ratio (97.4%), and the latency rate (10.9%) when compared to other popular models.

## 1. Introduction

The Internet of Things (IoT) is an important growing technology in the modern world, and its widespread adoption has contributed to exponential increases in overall network traffic [1]. IoT devices collect data in different applications, such as transportation networks, smart cities, healthcare applications, industrial environments, and commercial applications [2]. The IoT-based information that is gathered is transmitted via the Internet for further data analysis. Cyber information sharing (CIS) must be followed during data transmission to manage data security and confidentiality. CIS is the process of transmitting information by maintaining its secrecy [3]. However, the data-sharing process encounters several difficulties because of the complex network topologies and the large volume of data. The main objective of CIS is applied in the business and organization process to manage security-related risks [4]. Sharing threat information while adhering to security, privacy, regulatory, and legal compliance standards requires that a company rely on the expertise of its employees and other entities, such as those users who participate in cyber threat information-sharing organizations [5]. IoT device data is typically uploaded through sensors and sent to a third-party service providers via conventional data-sharing protocols [6]. Therefore, the third-party service providers incorporate statistical analysis and machine learning techniques to improve the data quality [7]. There is a trade-off between the benefits of individualized and high-quality service and the potential exposure of private information [8].

Recently, blockchain has provided an alternative route to increased security that is less explored and far less appealing to cybercriminals [9]. The blockchain method is more efficient at confirming data ownership and integrity, reducing vulnerabilities, and providing robust encryption [10]. Controlling unauthorized persons’ access to and use of a company’s data and resources is critical for keeping that data secure [11]. Access control rules do this by verifying users and granting them the permissions they need to access secure areas of the network and data [12]. Access control restricts an unauthorized person’s ability to view data by ensuring that only authorized people can access sensitive corporate information [13]. The data should be encrypted before being sent to the cloud servers so that, if other precautions fail, attackers can only see the data in its encrypted forms [14]. Encrypting the transmitted data using a secret key will ensure data security and confidentiality [15]. Traditional encryption methods may be employed, with the data owner sending the decryption key to the specified users in advance [16]. Symmetric encryption requires the data owner and users to either share the same key or agree on a shared key to decrypt the information [17]. Data owners cannot predict which user accesses their information; sensitive data must first be encrypted by utilizing a key that is only known to the data owner and then re-encrypted by utilizing a key known only to the data users [18]. A data owner must be online constantly to use this decrypt-and-encrypt system, which is problematic when there are more data elements and more types of data owners and consumers; thus, the issue’s complexity increases [19].

Classification schemes, or taxonomies, give techniques to comprehend the similarities and differences among objects under investigation and are thus often used in the cybersecurity information systems field because of their complexity. Regarding cybersecurity, a risk taxonomy may be an invaluable resource for identifying organizations’ perceived risks. CIS has been crucial in establishing this preventative measure by disseminating information about cyber threats to other organizations. To prepare for and prevent cyber-attacks, businesses can use CIS to identify emerging risks earlier. Although businesses must work together to share information about cybersecurity risks, one of the primary issues is doing so without compromising individual privacy. In addition, blockchain has proven useful in resolving the unreliability of data-sharing services, as it provides transparency, and no entities can deny it once they contribute to the system [20]. Blockchain provides another way to improve security, which is less well-known and far less hospitable to hackers. This method improves security, offers robust encryption, and confirms ownership and the integrity of data with more granularity. Data immutability is one of the blockchain’s major drawbacks in data security. A properly balanced distribution of network nodes is necessary for immutability to exist. The whole blockchain is at risk if a single organization controls more than 50% of the network’s nodes.

The main contributions of this paper are as follows:It designs the Access Control Enabled Blockchain (ACE-BC) framework to enhance the overall data security in CIS.It introduces the attribute encryption technique for public key encryption, therby allowing users to decrypt and encrypt messages based on user attributes.The numerical outcome was performed, and the recommended ACE-BC framework increased the throughput, data confidentiality, low latency, and computation time when compared to other models.

The rest of the paper is arranged as follows: Section 2 deliberates the literature review, Section 3 proposes the ACE-BC framework, Section 4 deliberates the results and discussion, and Section 5 concludes the research paper.

## 2. Literature Study

Randhir Kumar [21] proposed the InterPlanetary File System and Blockchain (IPFS-B) for secured distributed detection in the industrial image and video data security. To identify cases of unauthorized usage of media files in various formats, the author employed a perceptual hash (pHash) method. Before any material can be uploaded to the IPFS, its pHash is calculated and compared to the values already stored in the blockchain. Experiment photos are from the Caltech 256 dataset. The media would be checked for tampering if its pHash value is too similar to previously recorded ones. The results demonstrated that blockchain technology provides the benefit of non-involvement of third parties and, consequently, avoids a single point of failure. The limitation of IPFS-B model performance is less than the performance compared to other models.

Bin Jia et al. [22] suggested a blockchain-enabled federated learning data protection aggregation scheme (BFLDPAS) in industrial IoT (IIoT). Differential privacy, homomorphic encryption-based distributed K-means clustering, differential privacy-based distributed random forests, and homomorphic encryption-based distributed AdaBoost are examples of techniques that allow for multiple layers of protection when exchanging data and models. The authors concluded with a thorough security analysis and integrated the approaches with blockchain and federated learning. Numerical outcomes illustrated that the proposed aggregation scheme and working mechanism achieved high performance in the selected indicators. Furthermore, this study had small positive influences on enhancing the secure interchange and sharing of industrial information.

David W. Chadwick et al. [23] recommended a cloud-edge-based data security architecture (CEDSA) for sharing and examining cyber threat information (CTI). Before releasing CTI data for analysis, its owner may choose a sanitization method and trust level suitable for the data from plain text via pseudonymization/anonymization to homomorphic encryption. As a bonus, this sanitization may be handled by edge devices or cloud service providers, depending on the degree to which an organization trusts the latter. The authors discussed the trust architecture, cloud-edge infrastructures, and the organization methodology to meet the most severe needs for secure CTI information exchange. Finally, the authors summarized the deployment and testing conducted through four pilot projects that verified the reliability of their infrastructure. A limitation of the proposed architecture is that all data protected objects shared for analysis have to have the same standard data sharing agreement policy encapsulated in them.

Bao Le Nguyen et al. [24] discussed the privacy-preserving blockchain technique (PPBT) for the reliable and secure sharing of IoT information. Using multi-kernel support vector machine training techniques in partial views of IoT data from different sources can provide safe ant colony optimization. The authors employed ECC to provide a privacy barrier for the ant colony optimization using the multi-kernel SVM secure learning procedure, thus making it both effective and accurate. IoT information was encoded and kept on distributed ledgers, and the authors of this research utilized blockchain techniques to build safe and trustworthy data exchange platforms across many data sources. Based on the security evaluation results, they determined that the variables of the proposed model were safe for data analysts and that the confidentiality of essential data from every data source was protected. The simulation results also demonstrated that the suggested model outperformed the other methods. However, the downloaded file was unusable, and, false data was stored on the system for concealed persistence.

Bingqing Yang [25] deliberated the proof-of-stake, proof-of-work, and secure hash (PoS-PoW-Hash) algorithm for internet data security business risks based on blockchain. The platform was first put through its paces in a business risk simulation with a small firm, and then its users, the company’s workers, were polled on their overall impression of the service. The results of the tests suggest that the company’s business risk was reduced by 5–10% thanks to the design of the sub-platform and that workers were generally pleased with it. Like the source management routing algorithm (SMRA) and rough set-based attribute reduction (RSAR) techniques, the algorithm’s signature was simulated under realistic settings. This method performed better than others, as the results demonstrate. This research paves a new way for avoiding potential threats to medium-sized and small businesses. However, this study did not employ blockchain technology efficiently in Internet data security management and risk avoidance in business processes.

Latif et al. [26] introduced blockchain artificial intelligence (AI), and software-defined networks (SDN) to manage data security in cyber-physical systems. The system aimed to manage the data security and energy factor while transmitting data in the network. During this process, a blockchain approach was incorporated to improve overall security. In addition, PoW was applied to maintain public and private peer-to-peer communication. Thus, the blockchain-enabled software-defined network managed the overall data security and privacy. However, the energy consumption and resource limitations of the IoT devices were not considered to assess the performance of this architecture.

Fazal et al. [27] recommended a decision support system to manage privacy while sharing massive amounts of data in a third-party environment. The authors used COVID-19 patients’ health information to analyze security. Initially, the blowfish algorithm was applied to encrypt the attributes, and pseudonymization was applied to mask the quasi-attributes and identity attributes. The encrypted data was then linked to improving overall data security. Ultimately, the process effectively minimized unauthorized activities. However, this work only focused on privacy for contact tracing, which is unsuitable when it requires guaranteeing privacy and data security.

Using mobile computing technologies, Kaushal et al. [28] analyzed medical applications to manage data privacy and secrecy. Their work used cutting-edge encryption techniques to protect data from third-party access. The data was processed using a normalization process that minimized irrelevant information. A principal component analysis was applied to reduce the feature dimension. Finally, kernel homomorphism was utilized to maximize overall security. In addition, spider monkey optimization and two-fish encryption were applied to the data, thus enhancing overall data security, confidentiality, and integrity. A drawback of this method was that patient welfare could not be monitored in real-time.

Zhang Wenhua et al. [29] stated that the evolution of medical care is moving into a new era with the creation of Health 5.0. Blockchain, as a technology solution, has decentralization, safe sharing, high privacy, and non-tampering, which presents a breakthrough for the current bottleneck in EHR privacy and security development from a fresh viewpoint. Safeguarding patient medical information against cyberattacks and protecting privacy with verified access is one of the healthcare industry’s most essential problems. While blockchain security is the cornerstone of healthcare growth, the future development of blockchain security can largely rest on technological applications, application extending, and monitoring models.

WALID EL-SHAFAI et al. [30] suggested the Genetic Encryption Algorithm (GEA) for data authentication. Initially, the GEA initiates its search from a population of templates, rather than a single template. Then, some mathematical operators exploit the first population to generate successive populations. Lastly, the crossover and mutation operations create the final cancelable biometric data templates. The suggested framework attained an average AROC value of 0.9998.

Fursan Thabit et al. [31] recommended the lightweight cryptographic algorithm to enhance cloud computing data security. There is a need for a key of the same length (16 bytes or 128 bits) as the algorithm’s block size (16 bytes). It takes cues from the feistal and replacement permutation architectural techniques for more encryption complexity. By using logical operations, the method accomplishes Shannon’s notion of dispersion and confusion (XNOR, XOR, swapping, and shifting). The secret key length and the number of turns may be adjusted freely. When compared to other popular cryptographic systems utilized in cloud computing, the proposed algorithm’s test findings demonstrated high security and significant improvements in cipher execution time and security forces.

Based on the survey, there are several problems with existing models in achieving high throughput ratios, data confidentiality, low latency, and computation time, such as BFLDPAS [22], CEDSA [23], and PPBT [24]. Hence, this paper proposes the ACE-BC framework to enhance overall data security in CIS.

## 3. Access Control Enabled Blockchain (ACE-BC) Framework

Privacy, security, and interoperability are significant in the conventional data-sharing domain. First, cyber data often includes personal information that must be protected. As a result, the loss of such information might have serious consequences for the credibility and bottom lines of affected businesses. Second, there is no mechanism for the decentralized exchange of cybersecurity information. Single-point-of-failure and DDoS assaults are two examples of a centralized framework’s security and resilience vulnerabilities. Third, managing access rights to private data or sharing private information is a significant difficulty in cybersecurity information sharing. This study proposed the ACE-BC framework with an attribute-based encryption mechanism for cybersecurity information sharing. A blockchain platform supports the management of cybersecurity certification and vulnerability information. As a result of its several appealing features and qualities, such as enabling decentralization and secrecy maintenance, blockchain protocols have been identified as having the ability to change the foundations of information technology. Blockchain is the optimal architectural solution to guarantee distributed transactions amongst all parties in a trustless setting.

Figure 1 shows the proposed ACE-BC framework. In our design, there is a separate edge gateway for each IoT service. Each gateway on the network’s edge connects to the consortium’s blockchain as peer nodes and communicates with the cloud using fifth generation wireless networks (5G). The gateway node acts as an orderer node in the consensus procedure if it satisfies the requirements. Our architecture for the consortium blockchain does not include IoT devices as peer nodes; instead, they are linked to the edge gateway in their domains, with which they exchange data about access control via the lightweight Message Queuing Telemetry Transport (MQTT) protocol. Connecting the edge gateway to the 5G networks and the cloud in millisecond-level latency is made possible by the 5G base station. Due to the presence of the base station, edge gateways may make certain and rapid use of cloud-based storage services, thus allowing it to swiftly and reliably report data that can be remotely monitored. Certification authority (CA) is required for edge gateways to join the consortium blockchain network. With 5G, edge gateways may access cloud-based storage services, thus lightening the data storage load on the blockchain. It also can receive data supplied by the edge gateway for remote monitoring and provide unified application services that are associated with production.

### 3.1. Step 1: System Initialization

Data security during network transit necessitates using encryption algorithms, which facilitate the transformation of plain text into encrypted text and back again. In most cases, sensitive information is protected from public scrutiny when encoded or encrypted. Data is locked (encrypted), so only the owner (or someone with the correct decryption key) can access it. Plaintext refers to the original unencrypted material, whereas ciphertext describes the encrypted version using a secret key. A higher level of security in data transmission between client applications and servers is possible thanks to the encryption algorithm parameter. The attribute-based encryption scheme runs setup algorithms to build B from the prime number orders of q bilinear groups $H_{1}$, which is denoted as h, and the respective bilinear mappings are $e:H_{1}\times H_{1}\to H_{2}$ and described as $G:\;1\;\left\{0,1\right\}\;H$. The attribute-based encryption scheme randomly chooses parameter $\beta \in Z_{q}$ and creates public keys using Equation (1):(1)$$PK_{B}=e{\left(h,h\right)}^{b}.MK_{B}=h^{b}$$

In Equation (1), $PK_{B}$ is denoted as the public key, and the key server chooses the parameter $\alpha \in Z_{q}$ arbitrarily to produce the public and private keys.
(2)$$PK_{K}=h^{\alpha }.MK_{K}=\alpha$$

In Equation (2), $PK_{K}$ is defined as the public key generated from the master key MK.

### 3.2. Step 2: Key Generation

The attribute-based encryption scheme and the key server run the key generation algorithms and then create user keys based on the additional homomorphic encryptions. The attribute-based encryption scheme creates $S=Enc\;\left(PK_{B},\beta \right)$ and directs it to key servers. The key servers choose a arbitrarily and generate $\delta \in Z_{q}$, create $U=\left(S⊕Enc\left(PK_{B},\;\delta \right)\right)⊗\alpha$, and send it to the business entity. The attribute-based encryption scheme decrypts U to get Y based on additional homomorphic encryption, as defined in Equation (3).
(3)$$Y=Dec\left(MK_{A},U\right)=\left(\beta +\delta \right)\alpha$$

The attribute encryption mechanism randomly selects prime numbers β and $\in Z_{q}$, and the computing of $Z=h^{Y/\tau }=h^{\left(\frac{\left(\beta +\delta \right)\alpha }{\tau }\right)}$ is then sent to key servers. The key server calculates $M=Z^{1/\alpha^{2}}=h^{\left(\beta +\alpha \right)/\tau \alpha }$ and sends it to the business entity. The business entity utilizes *τ* to produce the user key $WK_{B}=D=M^{\tau }=h^{\left(\beta +\delta \right)/\alpha }$ and send it to the users securely. The key servers then run the key generation algorithms, arbitrarily choose $\delta_{i}\cup Z_{q}\;$ for every attribute of users $\beta_{i}\in W$, and produce and store attribute keys $WK_{K}$. The user’s attribute private key WK is self-possessed of user keys $WK_{B}$ and attribute keys $WK_{K}$ using Equation (4):(4)$$WK_{K}=\left({\left\{D_{i}=g^{\delta }G{\left(i\right)}^{\delta_{i}},D_{i}^{\text{′}}=h^{\delta_{i}}\right\}}_{i\in W}\right)$$

After generating the key values, the collected data is encrypted to manage the data’s security and confidentiality. The detailed encryption process is described next.

### 3.3. Step 3: Encryption

The data owners run the encoding algorithms, set the access policy tree T, encode the information N, and output the ciphertexts CT. First, data owners choose $DK\in \;Z_{q}$ arbitrarily and utilize DK to encode the data N by relying on symmetric encryptions. They then build the access strategy tree T, describe $Bk_{y}-1$ degree polynomials $p_{y}$ for every node y in the tree in B top-down way, and choose $w\in Z_{q}$ arbitrarily. For root nodes R of tree T, they describe $p_{R}\left(0\right)=w$. For other nodes y of tree T, they denote $p_{y}\left(0\right)=p_{parent}\left(index\left(y\right)\right)$ and choose the random variable to complete the description of $p_{y}$. Supposing that X signifies the set of attributes respective to the leaf node in the access policy tree T, ciphertexts are built by using Equations (5) and (6): (5)$$CT=\left(T,E=WEnc_{DK}\left(N\right),C=DK\cdot e{\left(h,h\right)}^{\beta y}\;,C=h^{\alpha w}\right),$$
(6)$${\left\{C_{x}=h^{p_{x}\left(0\right)},C_{x}^{’}=G{\left(Attribute_{x}\right)}^{p_{x}\left(0\right)}\right\}}_{x\in X}$$

The encrypted data is stored in the cloud to manage privacy and security. First, the request message is transmitted to the central authority to verify the user’s identity if the user requests detailed data. Once the user credentials are verified, the access control list is checked to improve overall data security and confidentiality. After verifying the user details in the access list, the decryption key is given to the user to access the data in the third-party server. The decryption process is described next.

### 3.4. Step 4: Decryption

The user requests decryption from the key server after receiving the ciphertext from loud service providers. The key servers run the model and decode the ciphertext by utilizing attribute keys. The procedure of decryption is established by recursive algorithms, which describe recursive algorithms $DecryptNode\;\left(CT,\;WK_{K},\;y\right)$, input ciphertexts CT, attribute keys $WK_{K}$, and nodes y in the access policy tree T. If y is a leaf node, express j=attribute y.
(7)$$DecryptNode\;\left(CT,\;WK_{K},\;y\right)=\frac{e\left(D_{j},C_{y}\right)}{e\left(D_{j}^{’},\;C_{y}^{’}\right)}=\frac{e\left(h^{\delta }G{\left(j\right)}^{\delta_{j}},h^{p_{y}\left(0\right)}\right)}{e\left(h^{\delta_{j}},G{\left(j\right)}^{p_{y}\left(0\right)}\right)}=e{\left(h,h\right)}^{\delta p_{y}\left(0\right)}$$

According to Equation (7) of the decryption progression, the user can decode $e{\left(h,h\right)}^{\delta w}\;$ only if the set of possessions that he owns fulfills the access strategy. The attribute encryption mechanism uses the arbitrary and distinctive attribute to create the private attribute key for every user, thus guaranteeing that the private attribute keys of every user are diverse. If collusive users can compute $e{\left(h,h\right)}^{\delta p_{y}\left(0\right)}$ from the respective node y but cannot compute $e{\left(h,h\right)}^{\delta y}$, they, therefore, cannot decrypt DK. The CIS process is illustrated in Figure 2.

Figure 2 shows the CIS model. The challenging tasks of internet networks are efficient data sharing and approved access control. Thus, the blockchain model is integrated with access control and cybersecurity information-sharing mechanisms to remove the problems in conventional policies. As a result, blockchain solves many more effective problems in providing data integrity, fairness, authenticity, security, and distribution. Smart contracts are utilized to handle access control and data sharing. Additionally, user behavior is monitored. Further, some permission stages are defined for the subject to access the object’s service. The edge nodes serve as a proxy by re-encrypting data for authorized users. By storing data at the network’s edge, edge devices can provide users with reliable and fast services. The data owner provides the re-encryption key, which is used to get the ciphertext from the cloud service providers and then decode it to reveal the user’s identity. The blockchain operates as the system’s trusted authority. The authorized organization issues private keys associated with individual identities. Distributed ledgers such as this improve data privacy and security by making transactions more trustworthy, public, and easy to verify. This allows data owners to better control their information. The blockchain system records the data’s owners and users and distributes membership keys. When a user requests data access, the owner uses the user’s credentials to create a new encryption key, which is then sent to the proxy server. The rules and restrictions for accessing and using the data are then sent to the blockchain. The identity of a data user is checked before authorizing access.

The trusted authority executes the setup procedure during the initial setup to provide basic system settings and a master key. Next, users’ keys are generated in real-time using the KeyGen method. The data’s owner then executes the encrypting algorithm to generate the ciphertext. The metadata is kept in a distributed ledger (blockchain), while cloud providers handle the ciphertext. In our architecture, using data caches as part of the forwarding process makes content delivery more resilient to packet losses, thereby increasing availability. Furthermore, the multipoint delivery system of an information-centric network guarantees the efficient usage of bandwidth and storage. In addition, bandwidth consumption decreases as the number of viewers grows, as the material will no longer be unicasted.

Figure 3 shows the flowchart of access right authorization. The ability might be produced by linking a virtual identity to permission, which would serve as information reflecting the right to access. As a result, the capacity to verify one’s identity is a key feature in avoiding forgery. In response to a user’s request for access, domain owners produce a capability token according to a determined access control strategy and initiate transactions to record the fresh token data in smart contracts. Capability pools on smart contracts organize many facets of identity, and these contracts can be verified and made consistent throughout the network of blockchain nodes. To determine whether or not to allow access to the service, service providers must first get the capability token from smart contracts utilizing the subject’s address and then make a decision based on the local access control strategy. Suppose the correct access validation is set up at local service providers. In that case, smart objects may participate in the access control decision-making process, thereby making it possible to provide a granular and adaptable service for controlling access to IoT devices. Our solution realizes the capability delegation mechanism by properly configuring a delegation set in the identification capability. After receiving a token update transaction from the user, the smart contract will verify the delegation right by examining the list of delegations. The user may delegate his or her capability tokens to another entity by attaching the target addressed to the delegatee, provided that the values of delegation depth are more than the count of components in the delegatee. A request to delegate capabilities is denied if this condition is not met. Each time a delegation transaction completes successfully, the queue delegatee grows by one item until it reaches the maximum length determined by the delegation depth. The revocation of capabilities considers two cases: removing a delegation’s rights and removing an individual’s capabilities. Only the domain owner can carry out a smart contract’s revocation activities in our proposed system. To cancel the delegation of authority for a certain entity, the domain owner may eliminate the addresses from the delegate by using the revocation of the delegation procedure. If the identity capability is revoked by setting the delegation depth to zero or removing the access right, the corresponding capability token is no longer usable by any previously linked entities.

Figure 4 illustrates blockchain-based key management. The operations provided are initialization registration, access record, access query, key updates, leaving a node, and revoking access. Topology, which is shared by all security access managers in the same deployment domain, is the foundation for verifying access query transactions. When adding a new user to a blockchain, they must first apply for initialization registration. The user entry has to stipulate its deployment tuple, encrypted access keys (for auditing), and approved node. Next, the subject should initiate an access query transaction to the security access managers to request access to the desired node. Security access managers running the blockchain have a complete picture of all authorized nodes on every device once the startup phase is completed, thus allowing them to more accurately determine and verify the legitimacy of access transactions. If a subject is allowed access to an object at some point throughout the key’s lifespan, it is up to the subject to determine whether or not to use that access. Therefore, the object’s access operation must be reported in the access record transaction with the subject’s access signature. It is at this stage that the access cycle often concludes. Finally, users must transmit a key update transaction to security access managers to record the key and its lifespan, which is necessary for security reasons, such as the eventual expiration of keys for previously allowed nodes. Typically, a node creates a transaction to announce an incoming action and lists its parents as the new parent nodes for its children just before it exits the network. Depending on its authority assignment mode, each node may choose whether or not to participate in this procedure. If a hacked or malicious node is identified, security access managers sign a novel transaction and add it to new blocks, thereby declaring the node’s security access revocation. As the blockchain connects all transactions related to a certain node, the revoked nodes cannot access (deny) or be accessible by the network (risk warning). The suggested ACE-BC framework increases the throughput ratio, data confidentiality, efficiency ratio, and low latency compared to existing systems.

## 4. Results and Discussion

Secure information exchange from IoT devices is illustrated together with the method of enforcing distributed access control. The created blockchain-based secure access control system was implemented using the NS2 simulation tool. Intel i3 CPU with 2GB RAM and the Ubuntu operating system were utilized during this process. In addition, 8GB RAM and an i7-4510U processor were utilized to develop the secure data transaction process. As data-sharing technologies have improved, they have slowly entered many businesses. Consequently, the usefulness of the data can only be realized via secure data exchanges. However, the original data-sharing architecture does not provide simple tracking of digital data use. The unwillingness of data suppliers to release their information is another issue. This study presented a strategy for sharing data that uses blockchain technology, thereby solving the security and control problems inherent in conventional centralized data sharing and administration. Furthermore, this research assessed the model’s usability and safety. This work provided a blockchain-based data sharing paradigm and demonstrated that it is practical, secure, easily governed, and highly efficient. This model was built on the ACE-BC framework, and the decentralized information security access was based on blockchain. In addition, the entire database was synchronously encrypted, which may prevent data content from leaking and function as the link of the entire architecture. It is recommended that the underlying technology be employed as distributed storage, which has the potential to overcome issues such as the existence of a single point of failure in centralized storage. The simulation parameters were related to the depiction of security specifications, which are defined in Table 1.

### 4.1. Data Confidentiality Ratio (%)

Blockchain provides a safe and effective platform for exchanging data. The suggested approach classifies users using label data to deliver more granular data sharing services to ensure data security. Data confidentiality is computed based on steps 1–4 in the proposed section (i.e., initialization, identity authentication, signature and verification, and data transfer) to accomplish a safe and efficient data exchange. The core of the information sharing system is the detection server. In this setup, a central server collects and processes label data from all clients, identifies communities using cosine similarity, and then publishes those communities on a blockchain. The blockchain client allows users to access shared analytics and collaborate on shared data. The suggested framework has been shown to provide significant performance improvements over existing methods in terms of time cost and throughput, as were measured in experimental settings. In addition, the efficiency of information exchange was significantly increased in experimental simulations of the suggested approach. Figure 5 illustrates the data confidentiality ratio of the proposed ACE-BC framework.

Figure 5 shows that the introduced ACE-BC framework attained a higher confidentiality ratio (97.54%) than the other methods. This is because successful public and private key generation helps encrypt the original data. In addition, access control mechanisms were utilized to maintain the data access restriction. Therefore, the ACE-BC approach improved the overall confidentiality rate compared to the BFLDPAS [22], CEDSA [23], and PPBT [24].

### 4.2. Throughput Ratio (%)

The throughput was computed by dividing the file size by the time to get the throughput in megabits, kilobits, or bits per second. Network throughput is the rate at which data may be sent from one location to another. It is common to practice expressing the speed of a network in terms of the number of bits transferred per second, such as megabits or gigabits. The inputs of this throughput analysis were data size and speed. The throughput metric was used to justify how effectively the information was securely transmitted to the end user. First, as an expensive cryptographic primitive, attribute-based encryption is too high to meet the excellent throughput requirement of time series IoT data in a business context. The approach suggested here used considerable decentralization to sidestep the issue of a single point of failure. First, the framework for this information exchange method was founded on blockchain technology and shared data storage devices. Blockchain and integrity protection are distributed and decentralized. Therefore, the failure of any one node will not compromise the system as a whole. Second, this paradigm allows for coordination between various attribute management authorities. By separating the roles of different attribute authorities, the system is more resilient to the failure of any of them and can better prevent illegal actions from disrupting service. Furthermore, unlike the alternative case, where the owner administers the attribute, this study decoupled attribute management from the data owner. By restricting the data owner to the data manager job and prohibiting them from doing any other user-managed actions, data unavailability is avoided due to the data owner’s inability to react promptly. The proposed model achieved the throughput ratio of 98.2%. Figure 6 shows the throughput ratio of the suggested ACE-BC framework.

### 4.3. Efficiency Ratio (%)

The efficiency of the blockchain network being computed includes throughput, latency, and scalability (i.e., the number of participants the blockchain network can serve). Efficiency is estimated by determining the ratio between the data transmitted securely and the total number of data items transmitted by the third party. Online data transfer provides clear benefits in terms of timeliness compared to the other transmission techniques. However, as more information becomes available online, there will be a greater need for safe data storage, sharing, and efficient data-sharing methods. The needs for privacy and security in today’s technologically advanced environment are difficult for standard data transmission methods to satisfy. Because of blockchain’s inherent decentralization, audibility, and tamper-proof nature, the suggested system is an intriguing candidate for research into next-generation data-sharing technologies. By combining blockchain technology with data sharing, secure and efficient data exchange may be achieved with only a few clicks. The suggested algorithm for method detection groups users into communities for sharing data based on their labels’ degree of similarity. The community detection results assessed by the sharing degree were used to determine the scope of data sharing, which can effectively reduce the volume of shared data queries and increase the efficiency of data sharing. The experimental findings validated the efficiency and safety of the suggested strategy for data exchange between clients. The proposed ACE-BC model attained an efficiency ratio of 97.4%. Figure 7 illustrates the efficiency ratio.

### 4.4. Latency Ratio

Network latency is calculated by the sum of all possible delays that a packet can face during secure data transmission. This study addressed existing problems by suggesting a blockchain-assisted distributed key-generation framework with cloud computing to decrease latency and multiple blockchains operated in the cloud to achieve domain access. To satisfy the IoT scenario’s low-latency and high-scalability necessities, this study introduced cloud managers to perform multiple blockchains composed of various blockchains in every deployment domain. The numerical outcomes demonstrated that network latency was the dominant factor in system effectiveness; thus, employing blockchain technology on the cloud, which is closer to the terminal device, was advantageous to IoT scenarios. The suggested key generation blockchain is activated on security access managers to deliver low latency key generation functions for users’ entries in similar arrangement domains. As a result, the suggested ACE-BC model achieved a lower latency rate of 10.9%. Figure 8 shows the latency ratio.

### 4.5. Computation Time

This study measured every user’s computation time when performing different encryption algorithms. This research observed that the computation time needed for blockchain operations accounted for much of the overall period. Notice that in this protocol, this study granted the data access key on blockchains. Therefore, the computation time did not vary with the data size. Furthermore, the period spent on blockchain operations accounted for most of every user’s operational period, since the receiver and sender had to complete verification to verify that they behaved fairly. In particular, the receiver only took 0.6 ms to determine his data access key in the blockchain operations. Figure 9 shows the computation time.

The suggested ACE-BC framework achieved a higher throughput ratio, data confidentiality, efficiency ratio, and computation time, as well as a lower latency than existing approaches, such as BFLDPAS [22], CEDSA [23], and PPBT [24].

## 5. Conclusions and Future Works

This paper presented the ACE-BC framework to enhance overall data security in CIS. Technology for storing and exchanging data has advanced to the point that it is now being implemented by many businesses. Therefore, the data’s full value can be tapped only via protected data transfer. On the contrary, the original data-sharing design does not allow for easy monitoring of digital data use. Meanwhile, data providers are often reluctant to provide their data, which is problematic. This research proposed a centralized data sharing and administration method that uses blockchain technology, thereby eliminating the security and control issues that previously plagued the field. The feasibility and security of the model were evaluated here. This study provided a data sharing paradigm that uses blockchain technology and showed that it is efficient, effective, and safe. Blockchain and the ACE-BC framework provided the foundation of this model’s decentralized, encrypted data storage and access system. In addition, the database was encrypted in real-time, which might prevent sensitive information from being leaked and serve as the glue that holds the whole infrastructure together. The experimental outcome illustrated that the recommended ACE-BC framework enhanced the data confidentiality ratio (97.54%), throughput ratio (98.2%), efficiency ratio (97.4%), and latency rate (10.9%) when compared to other popular models. The working principle was primarily used for distributed storage due to limitations associated with the centralized server and a single point of failure. In the future, functions like ciphertext search and policy hiding can be added based on the existing attribute-based encryption to meet more detailed access control needs.

## Figures and Tables

**Figure 1 sensors-23-03020-f001:**
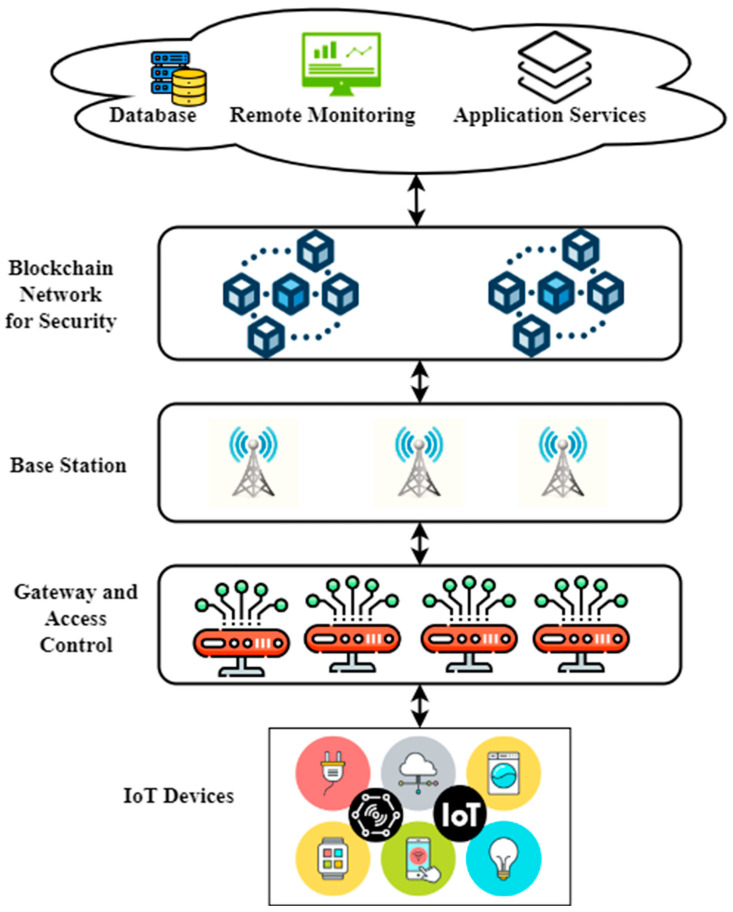
Proposed ACE-BC framework.

**Figure 2 sensors-23-03020-f002:**
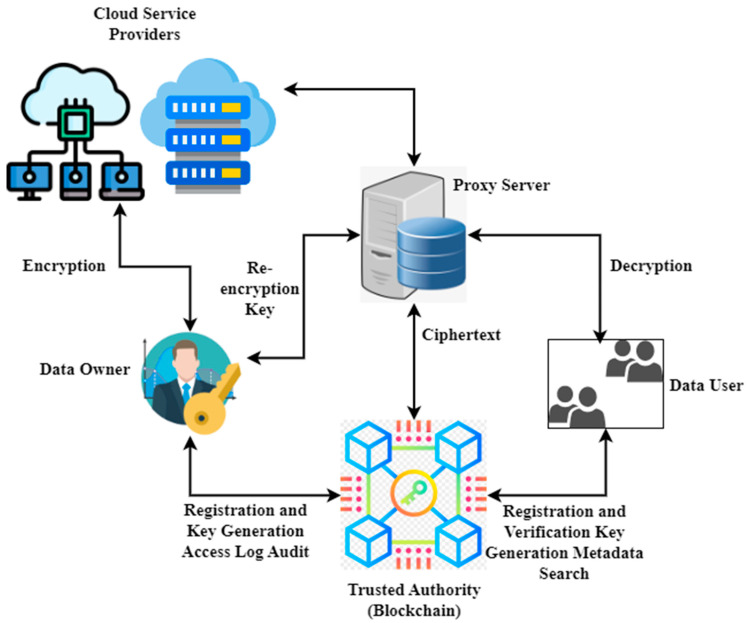
Cybersecurity information sharing (CIS) model.

**Figure 3 sensors-23-03020-f003:**
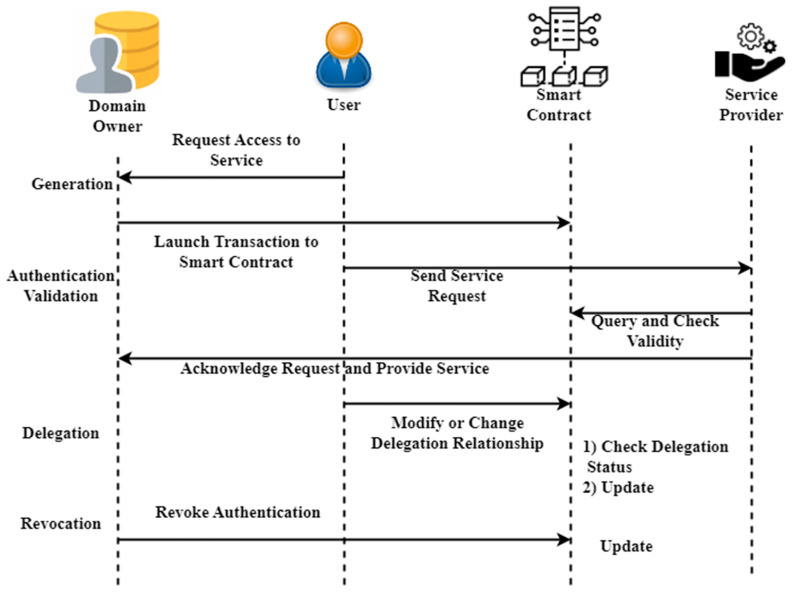
Flowchart of access right authorization.

**Figure 4 sensors-23-03020-f004:**
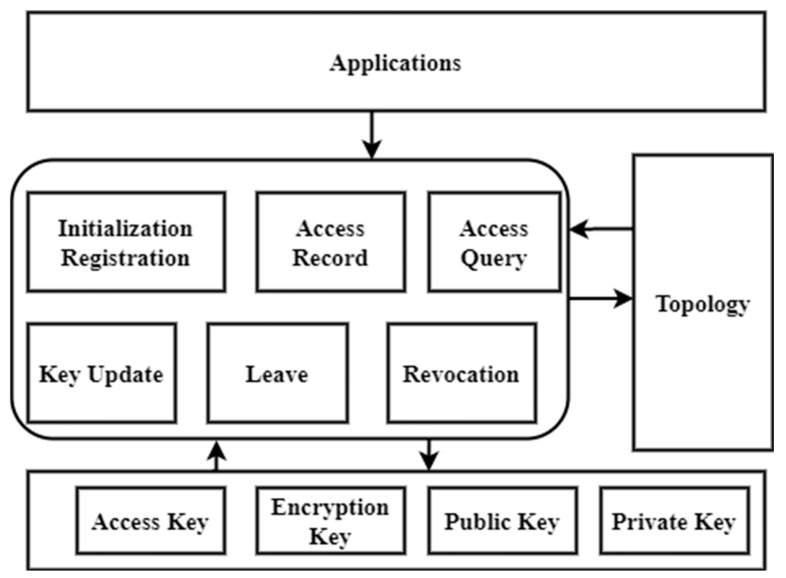
Blockchain-assisted key management.

**Figure 5 sensors-23-03020-f005:**
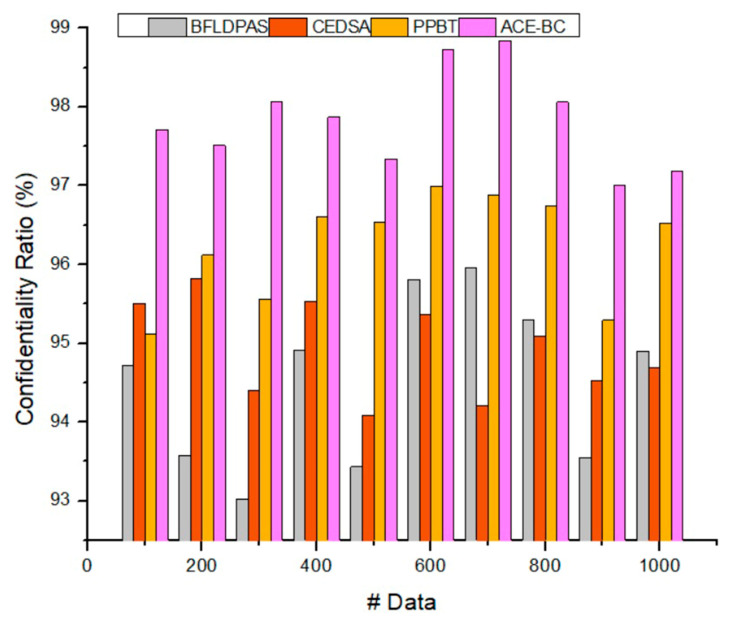
Data confidentiality ratio.

**Figure 6 sensors-23-03020-f006:**
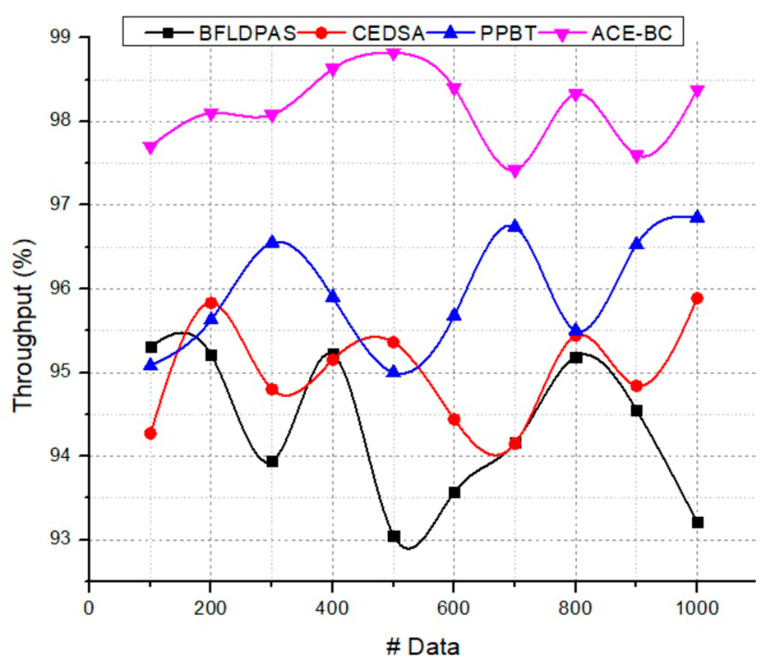
Throughput ratio.

**Figure 7 sensors-23-03020-f007:**
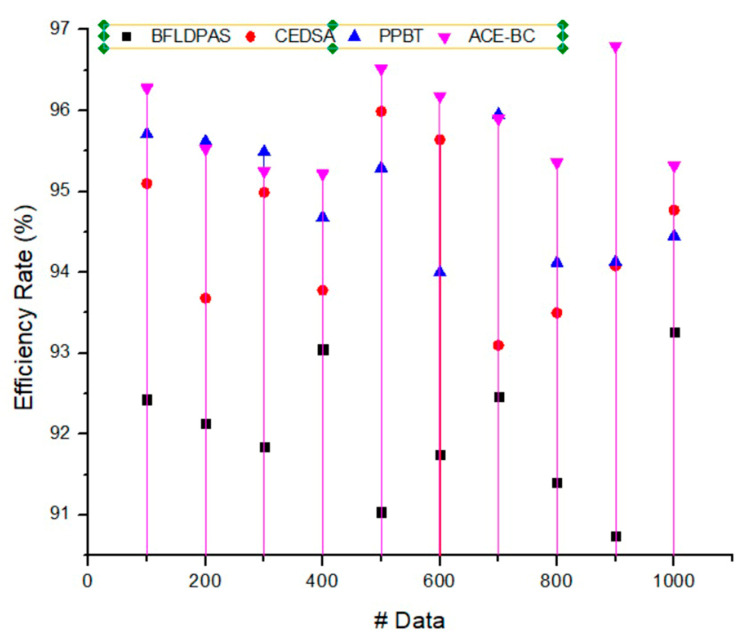
Efficiency ratio.

**Figure 8 sensors-23-03020-f008:**
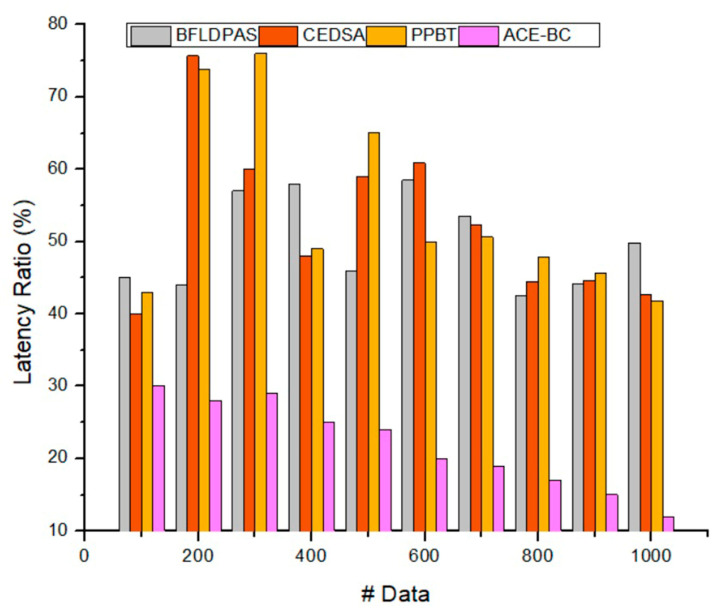
Latency ratio.

**Figure 9 sensors-23-03020-f009:**
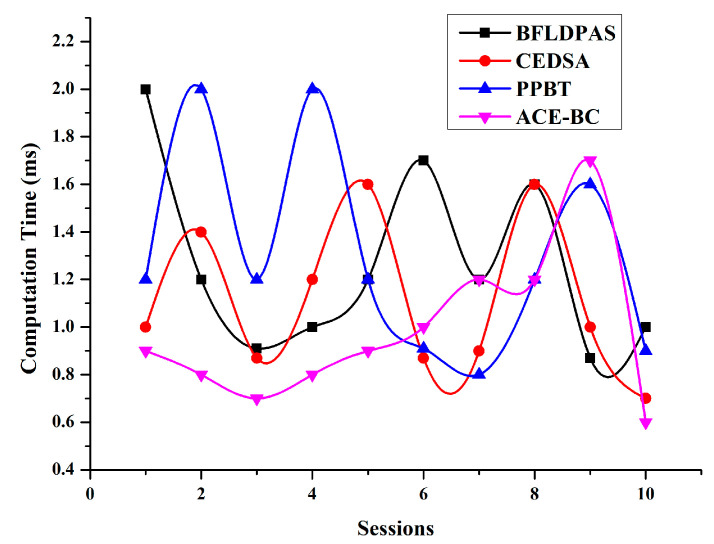
Computation Time.

**Table 1 sensors-23-03020-t001:** Simulation Parameters.

Parameter	Description
Block size	1 MB
Transmission Range	300 m
Message Size	1.5 GB
Total Number of Nodes	Random (10,000–30,000)
Number of Blocks	Random (100–250)
Security Protocols	AP, CKP
Traffic Type	Constant Bit Rate
Timestamp	128 bit

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
