# Peer review of "Applying Access Control Enabled Blockchain (ACE-BC) Framework to Manage Data Security in the CIS System"

_sensors, 2023, doi:10.3390/s23063020_

Round 1
Reviewer 1 Report
Though the author(s) claimed that their proposed approach was new, the kernel techniques used to establish the framework were developed a long time ago. The novelty of this paper is thus very limited as the author(s)’ efforts were mainly focused on combining these techniques. Authors have failed to convince that the paper brings a significantly new direction or novel idea in the field of cybersecurity information sharing compared to what has been already published; the contribution is very marginal.
The authors are recommended to improve the literature survey with some recent works.
- Discuss literature reviews based on their performance metrics along with the boons and limitations of each work surveyed.
-Discuss the reasons why the proposed work is offering better results in comparative analysis with the other similar methods.
- What are the strongest and weakest aspects of this work?
- About the datasets used by the authors, it is not clear if they are publicly available, otherwise (if they are private datasets) the possibility of reproducing the experiments, an indispensable aspect in any scientific work, would not be guaranteed, then the authors should clarify this aspect;
- The "Conclusions and Future Work" section appears excessively concise, then the authors should expand it by recapping all the steps to the proposed work, as to offer a brief but complete summary of it to the readers.
- While discussing results, it is interesting to justify the obtained results by mentioning advantages and disadvantages of the compared techniques, which is absent in this paper.
- The obtained experimental results must be more deeply discussed and justified. The Results and Discussion section must be enriched.
- The mathematical notation is not clearly explained, and somewhat confusing; several symbols and expressions remain unclear.
- What is the computation time for the algorithm?. Provide running time for the proposed method? Provide the comparison of computation time between the proposed method and other works.
- There are many algorithm parameters in the proposed method. What's the influence of these parameters?
- I suggest that the authors introduce certain taxonomy, at least through subsections.
Author Response
Though the author(s) claimed that their proposed approach was new, the kernel techniques used to establish the framework were developed a long time ago. The novelty of this paper is thus very limited as the author(s)’ efforts were mainly focused on combining these techniques. Authors have failed to convince that the paper brings a significantly new direction or novel idea in the field of cybersecurity information sharing compared to what has been already published; the contribution is very marginal.
-The authors are recommended to improve the literature survey with some recent works.
Ans: Thank you for your constructive comment. I have added some recent works as follows: WALID EL-SHAFAI et al. [30] suggested the Genetic Encryption Algorithm (GEA) for data authentication. Initially, the GA initiates its search from a population of templates, not a single template. Then, some mathematical operators exploit the first population to generate successive populations. Lastly, the crossover and mutation operations create the final cancelable biometric data templates. The suggested framework attains an average AROC value of 0.9998.
Fursan Thabit et al. [31] recommended the lightweight cryptographic algorithm to enhance cloud computing data security. There is a need for a key of the same length (16 bytes or 128 bits) as the algorithm's block size (16 bytes). It takes cues from the feistal and replacement permutation architectural techniques for more encryption complexity. Using logical operations like or, the method accomplishes Shannon's notion of dispersion and confusion (XNOR, XOR, swapping, and shifting). The secret key length and the number of turns may be adjusted freely. Compared to other popular cryptographic systems utilized in cloud computing, the proposed algorithm's testing findings demonstrated high security and significant improvements in cipher execution time and security forces.
- Discuss literature reviews based on their performance metrics along with the boons and limitations of each work surveyed.
Ans: Thank you for your constructive comment. Each work's performance metrics, advantages, and limitations have been surveyed.
-Discuss the reasons why the proposed work is offering better results in comparative analysis with the other similar methods.
Ans: Thank you for your constructive comment. Based on the survey, there are several problems with existing models in achieving high throughput ratios, data confidentiality, low latency, and computation time, such as BFLDPAS [22], CEDSA [23], and PPBT [24]. Hence, this paper proposes the ACE-BC framework to enhance overall data security in CIS.
- What are the strongest and weakest aspects of this work?
Ans: Thank you for your comment. Blockchain provides another way to improve security, which is less well-known and far less hospitable to hackers. This method improves security, offers robust encryption, and confirms ownership and integrity of data with more granularity. Data immutability is one of blockchain's major drawbacks in data security. A properly balanced distribution of network nodes is necessary for immutability to exist. The whole blockchain is at risk if a single organisation controls more than 50% of the network's nodes.
- About the datasets used by the authors, it is not clear if they are publicly available, otherwise (if they are private datasets) the possibility of reproducing the experiments, an indispensable aspect in any scientific work, would not be guaranteed, then the authors should clarify this aspect;
Ans: Thank you for your comment. No dataset is used in this study. The created blockchain-based secure access control system was implemented using the NS2 simulation tool. Intel i3 CPU, 2GB RAM, and the Ubuntu operating system were utilized during this process. In addition, 8GB RAM and an i7-4510U processor were utilized to develop the secure data transaction process.
- The "Conclusions and Future Work" section appears excessively concise, then the authors should expand it by recapping all the steps to the proposed work, as to offer a brief but complete summary of it to the readers.
Ans: Thank you for your constructive comment. The working principle is primarily used for distributed storage due to limitations associated with the centralized server and a single point of failure. In the future, functions like ciphertext search and policy hiding can be added based on the existing attribute-based encryption to meet more detailed access control needs.
- While discussing results, it is interesting to justify the obtained results by mentioning advantages and disadvantages of the compared techniques, which is absent in this paper.
Ans: Thank you for your comment. The suggested ACE-BC framework achieves a higher throughput ratio, data confidentiality, and efficiency ratio and computation time and lower latency than existing approaches, such as BFLDPAS [22], CEDSA [23], and PPBT [24].
- The obtained experimental results must be more deeply discussed and justified. The Results and Discussion section must be enriched.
Ans: Thank you for your constructive comment. The experimental results have been discussed.
- The mathematical notation is not clearly explained, and somewhat confusing; several symbols and expressions remain unclear.
Ans: Thank you for your constructive comment. Mathematical notation has been checked and verified.
- What is the computation time for the algorithm?. Provide running time for the proposed method? Provide the comparison of computation time between the proposed method and other works.
Ans: Thank you for your constructive comment. This study measures every user's computation time when performing different encryption algorithms. This research observes that the computation time needed for blockchain operations accounted for much of the overall period. Notice that in this protocol, this study grants the data access key on blockchains. Therefore, the computation time does not vary with the data size. Furthermore, the period spent on blockchain operations accounts for most of every user’s operational period since the receiver and sender must complete verification to verify they behave fairly. Especially the receiver only takes 0.6 ms to determine his data access key in the blockchain operations. Figure 9 shows the computation time.
- There are many algorithm parameters in the proposed method. What's the influence of these parameters?
Ans: Thank you for your constructive comment. Data security during network transit necessitates using encryption algorithms, which facilitate the transformation of plain text into encrypted text and back again. In most cases, sensitive information is protected from public scrutiny when encoded or encrypted. Data is locked (encrypted), so only the owner (or someone with the correct decryption key) can access it. Plaintext refers to the original unencrypted material, whereas ciphertext describes the encrypted version using a secret key. A higher level of security in data transmission between client applications and servers is possible thanks to the encryption algorithm parameter.
- I suggest that the authors introduce certain taxonomy, at least through subsections.
Ans: Thank you for your constructive comment. Classification schemes, or taxonomies, give techniques to comprehend the similarities and differences among objects under investigation and are thus often used in the cybersecurity information systems field due to their complexity. Regarding cybersecurity, a risk taxonomy may be an invaluable resource for identifying organizations' perceived risks.
I would like to thank Reviewer 1 for his time and suggestions. Hope the changes made in the revised manuscript can get his approval.

Reviewer 2 Report
This study focusing his work uses the Access Control Enabled Blockchain (ACE-BC) framework to enhance overall data security in Cybersecurity information sharing. Overall, the paper is good, however, there are few minor issues that must be addressed before the publication.
1. The CIS model need more details explanation.
2. Figures qualities are low. Improve all figures, some are blurry.
3. Author can look into some latest research on blockchains such as “Blockchain Technology: Blockchain Technology: Security Issues, Healthcare Applications, Challenges and Future Trends ” (Electronics, 2023).
4. Simulation parameters need to discuss.
5. References formatting should follow the same text style as body. Check the overall formatting of MDPI journal.
6. There are many instances where the language interferes with comprehensibility. The author should proofread and edit their manuscript for clarity.
Author Response
This study focusing his work uses the Access Control Enabled Blockchain (ACE-BC) framework to enhance overall data security in Cybersecurity information sharing. Overall, the paper is good, however, there are few minor issues that must be addressed before the publication.
- The CIS model needs more details explanations.
Ans: Thank you for your constructive comment. The challenging tasks of internet networks are efficient data sharing and approved access control. Thus, the blockchain model is integrated with access control and cybersecurity information-sharing mechanisms to remove the problems in conventional policies. As a result, blockchain solves many more effective problems in providing data integrity, fairness, authenticity, security, and distribution. Smart contracts are utilized to handle access control and data sharing. Additionally, user behavior is monitored. Further, some permission stages are defined for the subject to access the object's service.
- Figures qualities are low. Improve all figures, some are blurry.
Ans: Thank you for your constructive comment. All figures quality have been improved
- Author can look into some latest research on blockchains such as “Blockchain Technology: Blockchain Technology: Security Issues, Healthcare Applications, Challenges and Future Trends ” (Electronics, 2023).
Ans: Thank you for your constructive comment. Zhang Wenhua et al. [29] stated the evolution of medical care is moving into a new era with the creation of Health 5.0. Blockchain, as a technology solution, has decentralization, safe sharing, high privacy and non-tampering, which presents a breakthrough for the current bottleneck of EHR privacy and security development fresh viewpoint. Safeguarding patient medical information against cyberattacks and protecting privacy with verified access is one of the healthcare industry's most essential problems. While blockchain security is the cornerstone of healthcare growth, the future development of blockchain security can largely rest in technological applications, application extending, and monitoring models.
- Simulation parameters need to discuss.
Ans: Thank you for your constructive comment. The simulation parameters are related to the depiction of security specifications, which are defined in Table 1.
Table 1: Simulation Parameters
|
Parameter |
Description |
|
Block size |
1MB |
|
Transmission Range |
300 meters |
|
Message Size |
1.5 GB |
|
Total Number of Nodes |
Random (10,000-30,000) |
|
Number of Blocks |
Random (100-250) |
|
Security Protocols |
AP, CKP |
|
Traffic Type |
Constant Bit Rate |
|
Timestamp |
128 bit |
- References formatting should follow the same text style as body. Check the overall formatting of MDPI journal.
Ans: Thank you for your comment Reference formatting has been modified
- There are many instances where the language interferes with comprehensibility. The author should proofread and edit their manuscript for clarity.
Ans: Thank you for your constructive comment. Proofreading is done
I would like to thank Reviewer 2 for his time and suggestions. Hope the changes made in the revised manuscript can get his approval.

Round 2
Reviewer 1 Report
The topic is worthy of investigation. The authors have responded to the queries well. I suggest acceptance of the manuscript.